# Using ephaptic coupling to estimate the synaptic cleft resistivity of the calyx of Held synapse

**Martijn C. Sierksma**, **J. Gerard G. Borst***

Department of Neuroscience, Erasmus MC, University Medical Center Rotterdam, Rotterdam, The Netherlands

* g.borst@erasmusmc.nl

**Data Availability Statement:** The source code and data used to produce the results and analyses presented in the figures of this manuscript are available from the Bitbucket Git repository: https://bitbucket.org/gborst/electronic_synapse_model.

## Abstract

At synapses, the pre- and postsynaptic cells get so close that currents entering the cleft do not flow exclusively along its conductance, $g_{cl}$. A prominent example is found in the calyx of Held synapse in the medial nucleus of the trapezoid body (MNTB), where the presynaptic action potential can be recorded in the postsynaptic cell in the form of a prespike. Here, we developed a theoretical framework for ephaptic coupling via the synaptic cleft, and we tested its predictions using the MNTB prespike recorded in voltage-clamp. The shape of the prespike is predicted to resemble either the first or the second derivative of the inverted presynaptic action potential if cleft currents dissipate either mostly capacitively or resistively, respectively. We found that the resistive dissipation scenario provided a better description of the prespike shape. Its size is predicted to scale with the fourth power of the radius of the synapse, explaining why intracellularly recorded prespikes are uncommon in the central nervous system. We show that presynaptic calcium currents also contribute to the prespike shape. This calcium prespike resembled the first derivative of the inverted calcium current, again as predicted by the resistive dissipation scenario. Using this calcium prespike, we obtained an estimate for $g_{cl}$ of ~1 μS. We demonstrate that, for a circular synapse geometry, such as in conventional boutons or the immature calyx of Held, $g_{cl}$ is scale-invariant and only defined by extracellular resistivity, which was ~75 Ωcm, and by cleft height. During development the calyx of Held develops fenestrations. We show that these fenestrations effectively minimize the cleft potentials generated by the adult action potential, which might otherwise interfere with calcium channel opening. We thus provide a quantitative account of the dissipation of currents by the synaptic cleft, which can be readily extrapolated to conventional, bouton-like synapses.

## Author summary

At chemical synapses two neurons are separated by a cleft, which is very narrow. As a result, the concentration of released neurotransmitter can rapidly peak, which is essential for fast synaptic transmission. At the same time, the currents that flow across the

**Funding:** The authors received no specific funding for this work.

**Competing interests:** The authors have declared that no competing interests exist.

membranes that face the synaptic cleft are also substantial, and a fraction of these currents will enter the other cell. We made an electronic model of the synaptic cleft, which indicated that if the shape and fraction of these currents are known, the electrical resistance of the synaptic cleft can be inferred. We tested several of the predictions of our model by comparing the membrane currents during a presynaptic action potential in a giant terminal, the calyx of Held, with the much smaller prespike evoked by these currents in the postsynaptic cell. We estimate the cleft resistance to be about 1 MΩ, which means that the changes in the cleft potential due to the membrane currents can become large enough to have an impact on voltage-dependent calcium channels controlling neurotransmitter release.

## Introduction

When sir Sherrington introduced the term 'synapse' [1], it was still unknown whether or not nerve impulses reach their target through direct electrical coupling [2,3]. The visualization by electron microscopy of the synaptic cleft, the narrow gap between an axon and its target, put this debate to rest in favor of transmitter-mediated neurotransmission [4–6]. At about the same time, the electrical circuitry of the synapse was outlined, which included a cleft leak conductance ($g_{cl}$) linking synaptic currents with the interstitial space [7]. Since then, the purported role of the synaptic cleft for synaptic transmission has gradually become more important. Its small volume not only allows a rapid buildup of released neurotransmitter, but protons [8,9], and potassium ions [10–12] may accumulate as well, whereas calcium ions may become partially depleted [13,14]. Interestingly, during transmission a cleft potential ($v_{cl}$) may arise that may alter the dwelling time of mobile, charged particles within the cleft [15,16], the distribution of ligand-gated ion channels [17], or the kinetics of voltage-gated channels in the cleft-facing membrane [18]. These features all stem from the small size of the synaptic cleft, which thus forms a barrier for particle movements [19].

Except for some specialized synapses, the contribution of $g_{cl}$ to synaptic transmission has largely been ignored or deemed irrelevant, but its importance will of course depend on its magnitude and on the size of the currents that flow within the cleft. Eccles and Jaeger [7] already indicated that the maximal cleft current will be limited by this conductance. $g_{cl}$ decreases as the path length $l_{path}$ to the extracellular space increases ($g_{cl} \sim A/(R_{ex} \, l_{path})$, where $R_{ex}$ is extracellular resistivity in Ω cm and $A$ the area through which the current flows (see also ref. [20]). Due to their much longer path length, it has been argued that giant synapses have a lower $g_{cl}$ than conventional synapses [21]. Remarkably, estimates of $g_{cl}$ of very different synapses vary over four orders of magnitude (range 1.4 nS to 40 μS) without a clear relation to synapse size [7,11,18,20,22–30]. Despite the remarkable progress in our understanding of the synaptic cleft, we need a better theoretical understanding of its conductive properties, which is ideally confirmed by experimental observations.

One of the largest synapses in the mammalian brain is the calyx of Held synapse, which spans about 25 μm [31,32]. Its large size matches its strength, with the maximal glutamatergic conductance exceeding 100 nS, and its fast presynaptic AP allowing firing rates >300 Hz [33]. Large currents, related to the excitatory postsynaptic current (EPSC) and capacitive currents, need to flow within the cleft through $g_{cl}$, and under these conditions $g_{cl}$ may limit synaptic transmission. A characteristic feature of this synapse is that the presynaptic AP can be detected in the postsynaptic recording (Fig 1). This hallmark of the calyx of Held synapse has been called the prespike [34]. It is probably due to ephaptic coupling of the presynaptic AP via the

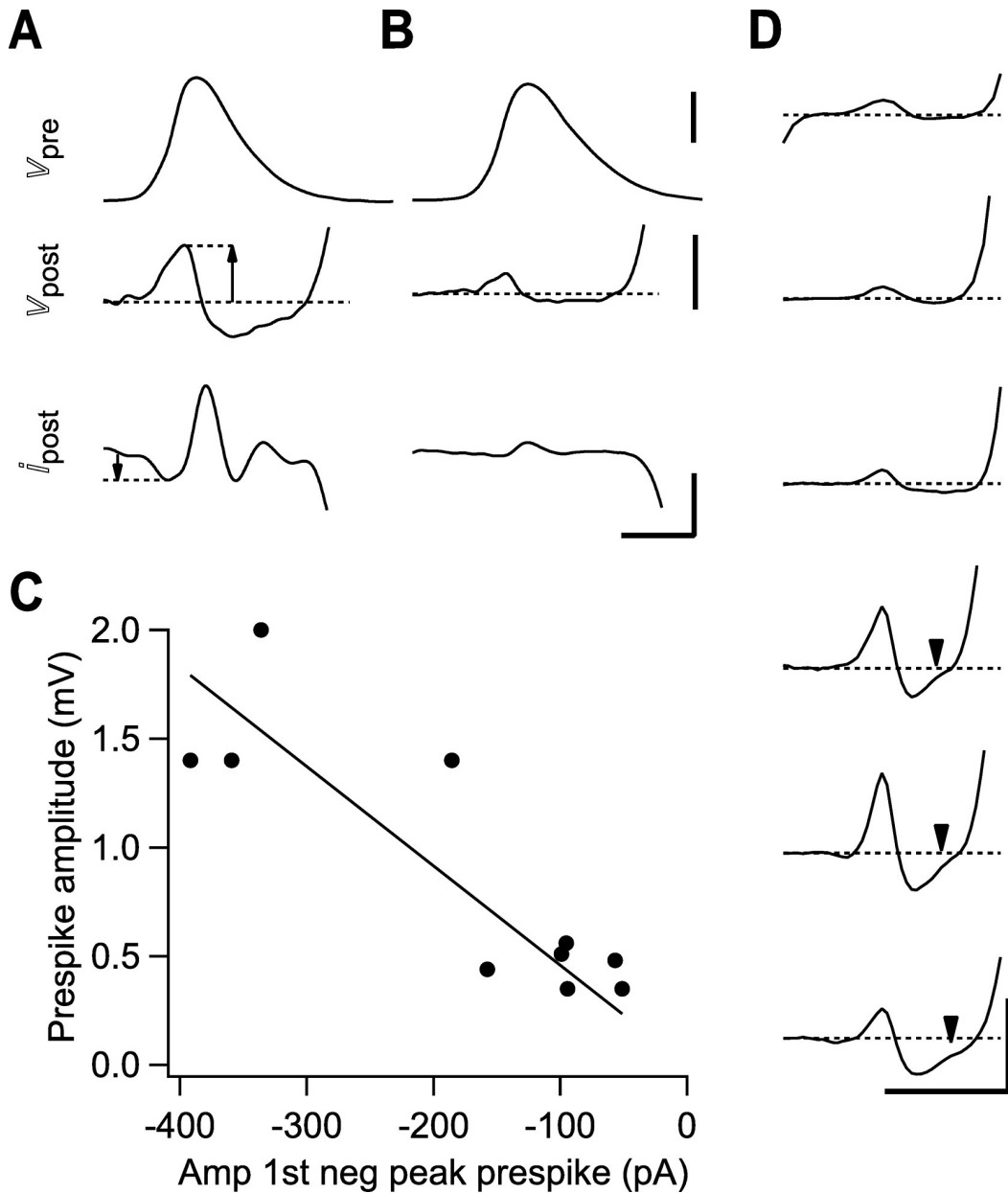

**Fig 1. Reduce by 25% Examples of prespikes.** (A-B) Two examples of prespikes in simultaneous pre- and postsynaptic patch-clamp recordings. From top to bottom: calyx of Held AP, which was elicited by afferent stimulation, CC prespike, associated VC prespike. Recordings were made at room temperature in brainstem slices from 7–10 days old Wistar rats [36]. Horizontal scale bar: 0.5 ms. Vertical (from top to bottom): 50 mV, 2 mV, 1 nA. Arrows indicate the amplitudes that were quantified in (C). (C) Relation between the amplitude of the positive peak of the prespike measured in CC and the amplitude of the first negative peak of the prespike measured in VC in the same cell. Slope of the fitted line is -4.6 mV/nA. (D). Six examples of CC prespikes from *in vivo* recordings of a target neuron of the calyx of Held in 3–5 days old Wistar rats [37]. During this period the calyx of Held expands over the postsynaptic cell. Note the small upward deflection immediately preceding the EPSP in the bottom three prespikes (arrow heads) which, we will argue, arises from the presynaptic calcium current running through the synaptic cleft (see also Fig 5B). Scale bar indicates 0.5 mV and 0.5 ms.

synaptic cleft [30,35], but the precise relation between the presynaptic AP and the prespike has not been studied analytically. Here, we use the prespike to infer some key electrical properties of this giant synapse. We provide a theoretical description of ephaptic coupling by defining the

electrical properties of the synaptic cleft, which is validated through dual patch-clamp recordings of the calyx of Held and its postsynaptic target.

## Results

The prespike is the ephaptically coupled calyx of Held AP recorded in its target neuron, a principal neuron in the medial nucleus of the trapezoid body (MNTB) [34]. Some current clamp (CC) and voltage clamp (VC) prespike examples are shown in Fig 1A and 1B. Fig 1C shows that the amplitudes of the CC and the VC prespikes recorded in the same cell are correlated. Fig 1D shows examples of the CC prespikes recorded during *in vivo* whole-cell recordings in young rats, whose shape resemble those recorded in the simultaneous pre- and postsynaptic slice recordings (Figs 1A, 1B and 2A).

### The prespike: a dissipation of cleft currents

The prespike is generated by currents that flow from the calyx to the principal neuron in the MNTB. While the current flow through the cleft depends on the synapse geometry, which we will consider later in this article, we can already get a good understanding by analyzing a prespike-generating, electrical circuit, similar to ref. [30], with the following components (Fig 2B): (1) the terminal's release face, consisting of voltage-gated conductances, a capacitor ($c_{\text{pre}}$), and a conductance ($g_{\text{pre}}$); (2) the synaptic cleft, which is separated from the reference by a leak conductance ($g_{\text{cl}}$), and (3) a passive neuron, which partly faces the synaptic cleft. This part consists of a capacitor ($c_{\text{syn}}$) and a leak conductance ($g_{\text{syn}}$). The remainder of the postsynaptic cell consists of a capacitor ($c_{\text{nsyn}} = c_{\text{post}}\text{-}c_{\text{syn}}$) and a leak conductance ($g_{\text{nsyn}} = g_{\text{post}}\text{-}g_{\text{syn}}$). As the prespike is small, we ignore the postsynaptic voltage-dependent conductances. We also did not include a direct resistive connection between calyx and target since there is no physiological evidence for gap junctions at the calyx of Held synapse [40], and capacitive currents are likely to dominate the electrical coupling: the susceptance at 1 kHz across an area of 1000 $\mu m^2$ is 63 nS ($2\pi f C_m A$, where $f$ is the frequency, $C_m$ is the specific capacitance, and $A$ the membrane area), which is much larger than the postsynaptic resting conductance of ~4 nS.

Using Kirchhoff's law of current conservation ($i_{\text{leave}} = i_{\text{enter}}$), we define the equation for $v_{\text{cl}}$ and subsequently the equations for the prespike (see S1 Appendix A1 for the derivations). The equation for $v_{\text{cl}}$ is $v_{cl}(g_{cl} + \sum g_{ion}) + (c_{pre} + c_{syn})dv_{cl}/dt = c_{pre}dv_{pre}/dt + \sum g_{ion}(v_{pre} - v_{ion})$ (S1 Appendix A1.5). The right-hand side shows the currents entering the cleft ($i_{\text{enter}}$), which are the presynaptic capacitive current and the currents through the voltage-dependent ion

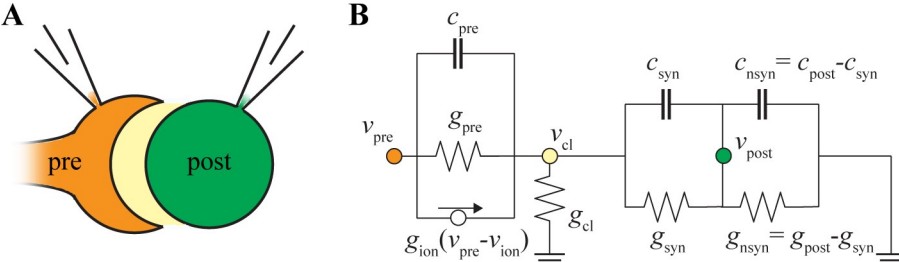

**Fig 2. An electrical circuit for the prespike.** (A) Drawing of the calyx of Held synapse. The calyx of Held terminal ('pre') covers a large part of the soma of its target neuron ('post'). In between is the synaptic cleft (not drawn to scale), a small space where membrane currents will flow during a presynaptic AP. (B) Electrical circuit to reproduce the prespike. Presynaptic membrane currents can leave the synaptic cleft via the cleft conductance ($g_{\text{cl}}$) or through the postsynaptic cell. It is assumed that the conductance from the edge of the cleft to ground is relatively large compared to the cleft conductance itself [38,39].

channels. The left-hand side of the equation shows the currents leaving the cleft ($i_{leave}$), which are generated by the cleft potential. This response will lie between two extremes: (1) the capacitive dissipation scenario, where the cleft poses a high resistance, resistive currents can be neglected, and the currents that leave the cleft are capacitive-only, and (2) the resistive dissipation scenario where the cleft is highly conductive, and the currents that leave the cleft are entirely resistive. For the capacitive dissipation scenario, we derive that $v_{cl} = v_{constant}+(c_{pre}+c_{syn})^{-1}\int i_{enter}dt$ (S1 Appendix A1.8); and $v_{cl}$ then resembles scaled, integrated $i_{enter}$, and therefore a scaled, presynaptic AP and the integrated voltage-gated currents. For the resistive dissipation scenario, we derive in Appendix A that $v_{cl} = i_{enter}(g_{cl})^{-1}$ (S1 Appendix A1.12); $v_{cl}$ will then resemble a scaled $i_{enter}$, which therefore should resemble the first derivative of the presynaptic AP.

The VC prespike represents the capacitive currents over the cleft-facing postsynaptic capacitance. In CC, the prespike currents will change the postsynaptic membrane potential. The equations for the currents entering and leaving the postsynaptic cell are $-i_{rec} = c_{syn}dv_{cl}/dt$ and $c_{post}dv_{post}/dt+g_{post}v_{post} = -i_{rec}$, respectively. The shape of the VC prespike is entirely determined by the shape of $dv_{cl}/dt$. We can derive how the VC prespike relates to $i_{enter}$ for the two extreme scenarios for the $v_{cl}$, For the capacitive dissipation scenario, we obtain $-i_{rec} = c_{syn}(c_{pre}+c_{syn})^{-1}i_{enter}$ (S1 Appendix A1.9), and therefore the VC prespike resembles the cleft currents scaled by the postsynaptic cleft-facing capacitance relative to the entire cleft-facing capacitance. Due to the giant size of the calyx of Held synapse, the presynaptic and postsynaptic capacitance facing the cleft are close to equal ($c_{syn} = c_{pre} = c_{cl}$) [41,42]. This means that: $-i_{rec} = 0.5\ i_{enter}$, and the other 'half' of $i_{enter}$ returns through the presynaptic membrane. (In other words, $v_{cl}$ reduces the capacitive currents from the presynaptic AP ($c_{pre}\ d(v_{pre}-v_{cl})/dt$) by half, which dissipate capacitively to the postsynaptic neuron.) For the resistive dissipation scenario, we obtain $-i_{rec} = c_{syn}(g_{cl})^{-1}di_{enter}/dt$ (S1 Appendix A1.13). For the calyx of Held, the ratio $c_{cl}/g_{cl}$ defines $\tau_{cl}$, the cleft time constant, and we can therefore write that $-i_{rec} = \tau_{cl}\ di_{enter}/dt$. The VC prespike then resembles the (inverted) second derivative of the presynaptic AP scaled by the constant $c_{syn}c_{pre}/g_{cl}$, which for the calyx of Held becomes $c_{cl}^2/g_{cl} = g_{cl}\tau_{cl}^2$. We therefore conclude that if the VC prespike resembles the (inverted) first derivative of the presynaptic AP, it indicates that cleft resistance is high, whereas if it resembles the (inverted) second derivative of the presynaptic AP it indicates a relatively low cleft resistance.

Does the VC prespike resemble the first or second derivative of the presynaptic AP? We analyzed simultaneous whole-cell presynaptic current clamp recordings with postsynaptic voltage clamp recordings [36,43]. In these recordings, the afferent axon of the calyx of Held was stimulated with an electrode to trigger a train of presynaptic APs, and the postsynaptic responses were recorded simultaneously (Fig 3A). The prespike generally matched the scaled, inverted second derivative of the AP (Fig 3B), taking into account that it may have been filtered by the series resistance. Fig 3C shows that the time course of the second derivative of the AP indeed provides a better match for the prespike than its first derivative ($n = 13$ paired recordings). This is in agreement with the idea that the cleft currents mainly dissipate resistively. During the train, the presynaptic AP becomes smaller and slower [36,44], resulting in a substantial reduction in the presynaptic capacitive current amplitudes. To investigate the impact of these changes on the prespike, we plotted the first two peaks of the prespike against the first two peaks of the first or the second derivative (Fig 3D). In each case, the second derivative of the presynaptic AP gave a better fit (Fig 3D-E, AP' versus AP": Pearson's $r = -0.967 \pm 0.018$ vs. $-0.979 \pm 0.018$, paired T-test, $t_{12} = 5.4$, $p = 0.00017$). Furthermore, this fit should cross through the origin, since either $-i_{post} = g_{cl}\tau_{cl}^2 d^2v_{pre}/dt$ or $-i_{post} = 0.5\ c_{pre}\ dv_{pre}/dt$ for the resistive dissipation or the capacitive dissipation scenarios, respectively. The absolute

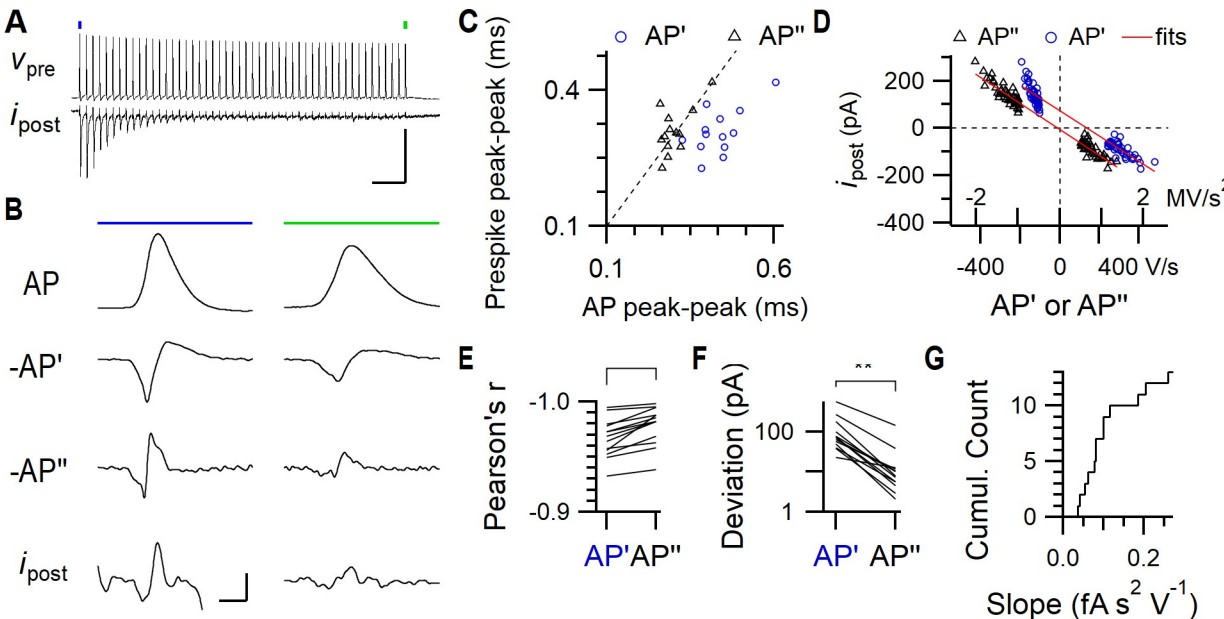

**Fig 3. Relation between the calyceal AP and the prespike.** (A) An example of a train of presynaptic APs ($v_{pre}$) elicited by afferent stimulation and the postsynaptic voltage-clamp recording ($i_{post}$). A large EPSC is elicited by the first AP, which rapidly depresses during the train. Stimulation artefacts were blanked. The first AP (blue dot) and the last AP (green dot) are shown on an expanded scale in B. Horizontal scale bar: 50 ms. Vertical scale bar: 90 mV, 1.5 nA. (B) The left and right column show the first and last AP of the train, respectively. *From top to bottom*: the AP recorded in CC from the calyx of Held, its inverted first (-AP') and second derivative (-AP''), and postsynaptic voltage clamp recording showing the accompanying prespike ($i_{post}$). Vertical scale bars (from top to bottom panels): 40 mV, 0.3 kV/s, 2.5 MV/s$^2$, 0.2 nA. Horizontal: 0.5 ms. (C) Relation between the delay between the first negative and positive peak of the prespike and the first two peaks in the first (blue, circle) and second (black, triangle) derivative of the presynaptic AP of the first AP-prespike pair in the train. Dashed line is the identity line. (D) Relation between the amplitude of the first negative and positive peak of the prespike and the first two peaks in the first (blue, circle) and second (black, triangle) derivative of the presynaptic AP of all AP-prespike pairs from an example train. The red lines indicate the linear regression lines. Note how the regression line crosses closer to the origin for the second derivative than for the first derivative. (E) Comparison for each train of Pearson's *r* for the regression lines. (F) Comparison of the absolute deviation at 0 V/s (AP') versus 0 V/s$^2$ (AP''). (G) Cumulative distribution of the regression slope. Paired T-tests: ** p<0.01, *** p<0.001.

deviation for the second derivative at 0 V/s$^2$ was significantly less for every recording than for the first derivative at 0 V/s (Fig 3F, AP' versus AP'': deviation = 122 ± 145 pA versus 20 ± 37 pA, paired T-test, $t_{12}$ = 3.3, $p$ = 0.0060). These fits also yielded an estimate for $g_{cl}\tau_{cl}^2$, namely the slope from the fit of the second derivative and the prespike amplitudes, which was 0.11 ± 0.07 fA s$^2$ V$^{-1}$ (Fig 3G). These observations are thus in agreement with the prespike resembling the inverted second derivative of the presynaptic AP, suggesting that the cleft currents mostly dissipate through the cleft conductance.

## The calcium prespike

Voltage-gated currents inside the cleft also contribute to $v_{cl}$. We explored how the presynaptic calcium current would affect the prespike (S1 Appendix A2 for the derivations). As the prespike depends on the current entering the cleft $i_{enter}$ irrespective of the origin of the current, a voltage-dependent calcium current in the cleft should also contribute to the prespike. We call the part of the VC prespike that is determined by the cleft calcium current, the calcium prespike. For the capacitive dissipation scenario, we expect that the calcium prespike is $-i_{rec} = i_{ca}$ (S1 Appendix A2.3). (Here, $i_{ca}$ is defined as the recorded current at the presynaptic terminal which is the net current through the calcium channels and the cleft-facing capacitance, and by definition, this current dissipates across the postsynaptic membrane.) For the resistive dissipation scenario, the calcium prespike is $-i_{rec} = c_{syn}(g_{cl})^{-1}di_{ca}/dt$, which we rewrite as $-i_{rec} = \tau_{cl}$

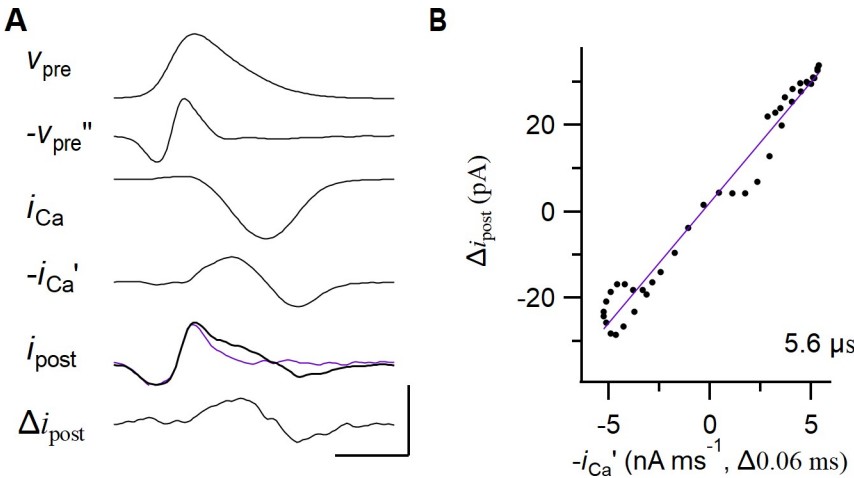

**Fig 4. Relation between the calcium current and the prespike.** (A) *from top to bottom*: The presynaptic voltage-clamp waveform ($v_{pre}$), the inverted second derivative of the waveform ($-v_{pre}''$), the presynaptic calcium current ($i_{Ca}$; after P/5 subtraction), its inverted first derivative ($-i_{Ca}'$), the active (black, thick) and passive prespike ($i_{post}$; magenta, P/5-scaled), and the calcium prespike which is the difference between the active and the passive prespike ($\Delta i_{post}$). The passive prespike is generated by injecting the presynaptic AP at 1/5th of its amplitude. While this generates passive currents across the presynaptic cleft-facing membrane, it does not elicit the presynaptic calcium current. Voltage-gated sodium and potassium currents were blocked (see Materials and Methods). Vertical scale bars: 110 mV, 7 MV/s², 2 nA, 15 nA/ms, 0.15 nA, 0.1 nA. Horizontal: 0.5 ms. (B) Relation between the inverted first derivative of the presynaptic calcium current and the calcium prespike (black dots). Maximum correlation was obtained by introducing a delay of 60 μs to the first derivative. The slope of the regression line (magenta) is shown in the bottom right corner.

$di_{ca}/dt$ (S1 Appendix A2.4). The calcium prespike will either resemble an inverted calcium current, or an inverted, scaled first derivative of the calcium current.

We turned to simultaneous pre- and postsynaptic whole-cell voltage clamp recordings of the calyx of Held synapse that were originally reported in ref. [44], which were used to evaluate the presence or absence of a calcium prespike. In these experiments the presynaptic compartment was voltage clamped with an action potential waveform (APW) command while the presynaptic calcium currents were pharmacologically isolated (Fig 4A). To isolate the calcium prespike from the recorded prespike, we averaged the postsynaptic responses to the P/5 AP waveforms and subtracted the upscaled, passive prespike from the 'active' prespike. The remaining currents indeed resembled the first derivative of the calcium current (S1 Appendix A2; Figs 4A, 4B and S1), confirming not only the existence of a calcium prespike, but also that its shape is in agreement with the cleft currents dissipating through the different, resistive routes.

From the analytical derivations for the resistive dissipation scenario, we know that the presynaptic calcium current and the calcium prespike are related by the cleft time constant $\tau_{cl}$. To estimate $\tau_{cl}$, we first analyzed the presynaptic calcium current that was evoked by the APW. It had a peak amplitude of -1.9 ± 0.6 nA and a full-width at half-maximum (FWHM) of 0.51 ± 0.05 ms ($n$ = 5 paired recordings). Next, we obtained the values for the calcium prespike. Its peak-to-peak amplitude was 80 ± 60 pA and its peak-to-peak delay was 0.54 ± 0.11 ms. This peak-to-peak delay should be about twice $\tau_{Ca}$, which was thus slightly higher than the 0.21 ms reported in ref. [45]. Its time course closely matched our prediction for the resistive dissipation scenario (Figs 4 and S1, S1 Appendix A2). The amplitude of the calcium current was linearly related to the calcium prespike peak ($r$ = 0.98, $p$ = 0.003); the calcium prespike peak was 1.9 ± 0.7% of the calcium current amplitude. In S1 Appendix A2 we derived that this percentage should be $\tau_{cl} \tau_{Ca}^{-1} e^{-0.5}$, yielding an estimated $\tau_{cl}$ of 9 ± 5 μs. As an alternative approach, we

plot the calcium prespike against the first derivative of the presynaptic calcium current. Its maximal slope (allowing for a slight time shift) provided a $\tau_{cl}$ of 5.7 ± 1.9 μs (Figs 4B and S1B). The two methods thus gave comparable values for the cleft time constant.

The ratio of cleft capacitance $c_{cl}$ and cleft leak conductance $g_{cl}$ constitutes $\tau_{cl}$. If we assume that $c_{cl}$ is in the order of 10 pF, the estimated $g_{cl}$ becomes about 0.5–1 μS. From the circuit analysis we know that the VC prespike coming from the presynaptic AP does not scale the same way to $c_{cl}$ and $g_{cl}$ as the calcium prespike from the presynaptic calcium current (*cf.* S1 Appendix A1 with A2). This difference allows us to further define these two parameters. From the calcium prespike we obtain $\tau_{cl}$. As the equation for the capacitive prespike can be written as $-i_{post} = g_{cl}\tau_{cl}^2 d^2 v_{pre}/dt$, we find that we can calculate $g_{cl}$. The first peak of the VC prespike had an amplitude of -120 ± 110 pA, which corresponds to the capacitive currents. We can then calculate a $g_{cl}$ of 2.6 ± 0.8 μS and a $c_{cl}$ of 24 ± 16 pF.

For $\tau_{cl}$ we assumed that the calcium current entirely originated from the membrane facing the cleft. These estimates should be considered as a lower bound as some of the current may not arise from the cleft-facing membrane. Immunolabeling for calcium channels identified a fraction of the N-type and R-type calcium channels to be in the non-cleft-facing membrane [46]. If we assume that only two-third of the calcium current originated from the cleft, our estimate of $\tau_{cl}$ increases to 13.5 ± 7.8 μs. This estimate could be considered as an upper bound, as less than 50% of the calcium current was mediated by N-type and R-type channels, and the other, P/Q-type, calcium channels were mostly associated with the release face [46]. When we assume that 2/3 of the total presynaptic calcium current originated from the cleft, we calculate a $g_{cl}$ of 1.2 ± 0.4 μS and a $c_{cl}$ of 16 ± 11 pF. With these values $g_{cl} \tau_{cl}^2$ was 0.07, 0.08, 0.11, 0.28 and 0.82 fA s$^2$ V$^{-1}$ (0.27 ± 0.32 fA s$^2$ V$^{-1}$), which is in agreement with Fig 3G. Based on our limited data set, we determine that $g_{cl}$ ranges between 1–2 μS.

## No evidence for a potassium or a sodium prespike

Besides the capacitive currents and the voltage-gated calcium channels, other voltage-gated channels facing the cleft may also contribute to the cleft currents. We therefore explored the potential contribution of Na$^+$ and K$^+$ currents to the prespike, the sodium and potassium prespike, respectively. We estimated their density based on literature values, and assumed a homogeneous distribution within the terminal. As voltage-gated channels in the cleft-facing membrane sense $v_{pre}$-$v_{cl}$ instead of $v_{pre}$, their gating depends on $v_{cl}$. This feedback effect of $v_{cl}$ could not be incorporated in our calculations. Instead, we therefore simulated the impact of voltage-gated currents on the prespike by numerically solving ordinary differential equations. Fig 5 illustrates that the presence of these channels at the release face may have a large impact on the shape and size of the prespike. Due to the differences in their gating properties, the timing of each current during the presynaptic AP is quite different (Fig 5A), and the same holds for their impact on $v_{cl}$ and on the prespike (Fig 5B). The prespike examples shown in Fig 1 did not resemble the prespike of the homogeneous model, which included Na$^+$, K$^+$ and Ca$^{2+}$ currents at the cleft-facing membrane, suggesting that some voltage-gated ion channels are excluded (at least functionally) from the release face. In particular, the voltage-gated sodium conductance generated a current that partially counteracted the first prespike peak. Such rapid changes in the prespike during the rising phase of the presynaptic AP would point at a contribution by sodium channels. However, consistent with previous observations that the terminal is mostly devoid of sodium channels [47–49], these fast changes were not observed in the prespike. We therefore conclude that there is no substantial contribution of sodium channels at the release face to the prespike.

Potassium channels do not start to open until around the peak of the presynaptic AP. They therefore do not impact the first part of the VC/CC prespike. The potassium current would

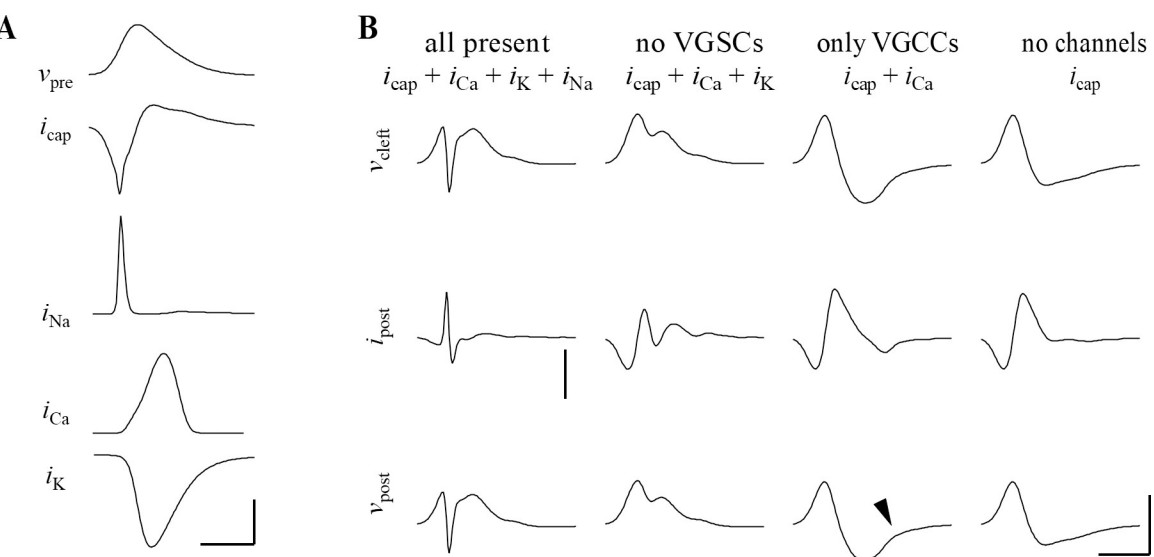

**Fig 5. Impact on the prespike of voltage-gated ion channels located at the release face.** (A) Timing of different voltage-gated currents at the release face during the AP. Note that the amplitudes of each of the currents depend on cleft potential, thus creating a mutual dependency. Vertical scale bar: 60 mV, 2.4 nA, 1 nA, 1 nA, 3 nA. Horizontal: 0.5 ms. (B) The cleft potential (top), the VC prespike (middle), and the CC prespike (bottom) in the presence or absence of different voltage-dependent ion channels at the cleft. All present: homogeneous density for each channel; no VGSCs: no sodium conductance at the release face; only VGCCs: both calcium and capacitive currents at the release face; no channels: the capacitive-only model. The conductances that are present in the model are indicated by their currents at the top. Arrow head indicates the small deflection in the CC prespike that is generated by the calcium prespike. Scale bars: 4 mV, 0.2 nA, 2 mV.; 0.5 ms, except for the homogeneous model where the vertical scale bar corresponds to 1.6 nA.

contribute to a more positive synaptic cleft potential, which is opposite to the effect of the calcium current. If the potassium current at the cleft were significantly larger than the other currents, we would expect a fast inward current in the VC prespike after the first positive peak, and the CC prespike would become a single positive deflection. However, CC prespikes are typically biphasic (Fig 1), therefore the contribution of the potassium current is unlikely to outweigh the contribution of the other currents.

Lastly, we revisit the impact of calcium channels on the prespike. The observation of a calcium prespike already demonstrated the presence of these channels at the release face. The calcium current occurs during the repolarization phase of the presynaptic AP [45], and therefore contributes little to the first two peaks of the VC or to the first peak of the CC prespike. Its contribution is easily obscured by noise or by the EPSC, which starts rapidly after the calcium-induced glutamate release. Nevertheless, it may generate a small, upward inflection in the CC prespike (Fig 5B) and such an inflection can sometimes be observed (Fig 1D).

These simulations show that the prespike waveform may contain information about the presence of voltage-dependent ion channels at the presynaptic membrane facing the cleft. This means that an experimental study of the impact of specific channel blockers on the prespike waveform may be informative about their presence at the release face.

## Voltage-dependent kinetics of cleft-facing calcium channels

During development the presynaptic AP narrows from a half width of 0.5 to 0.2 ms [50]. The briefer AP necessitates a much larger capacitive current, which, all else being equal, would generate a larger change in $v_{cl}$, which may impact the calcium channel kinetics. We again simulated the impact of $v_{cl}$ on the calcium current by solving ordinary differential equations for different values of $g_{cl}$ (Fig 6). For a slow AP with a $g_{cl}$ of 0.67 µS and a $c_{cl}$ of 10 pF, $v_{cl}$ first rises

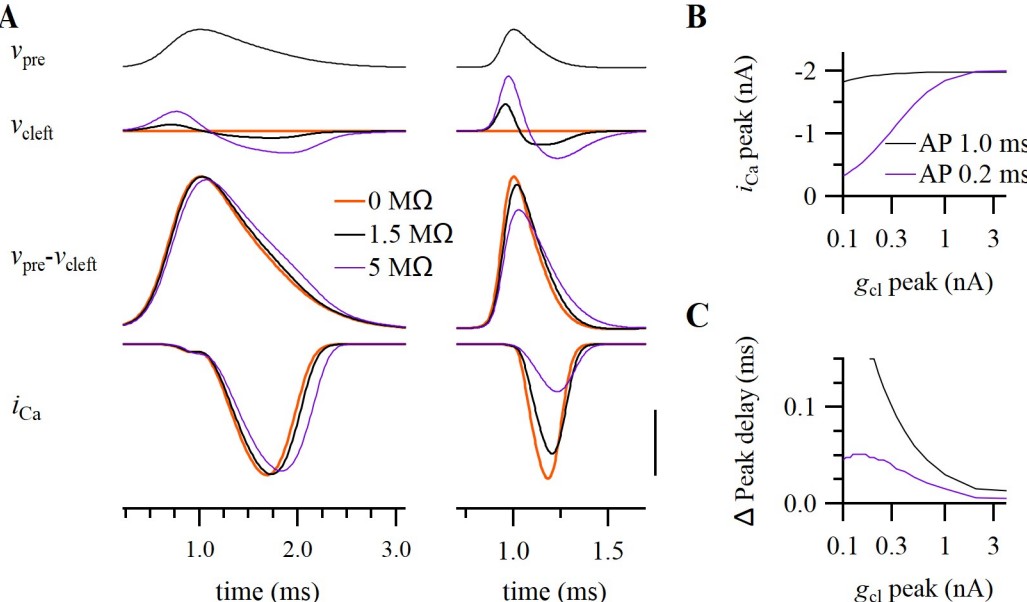

**Fig 6. Potential impact of the cleft potential on presynaptic calcium currents.** (A) Simulation of the impact of a slow and a fast AP (top; FWHM: left 1.0, right 0.2 ms) on the cleft potential (second row), the transmembrane potential sensed by the calcium channels in the cleft (third row) and the presynaptic calcium current (bottom) at three leak resistances (1/$g_{cl}$: 0 MΩ, 1 MΩ and 5 MΩ). Scale bar: 200 mV, 40 mV, 50 mV, 1 nA. (B) Relation between peak calcium currents and leak resistance for the two APs. Driving force slightly increases (not illustrated). (C) The change in delay between the calcium peak current and the AP for the two APs (solid line). The dashed line shows the delay after correcting for the decrease in the calcium current shown in B. Calcium conductance density was adjusted to 0.2 nS μm$^{-2}$ and 2.8 nS μm$^{-2}$ for the slow and fast AP, respectively, in order to elicit a 2 nA-current at 0 MΩ cleft leak resistance. The apposition area was 1000 μm$^2$. Other conductances were set to 0 nS.

to a maximum of +3.9 mV, and then decreases to a minimum of -4.2 mV. While $v_{cl}$ has little effect on the current amplitude (Fig 6B), it generates a delay of 45 μs in the onset of the calcium current (Fig 6C). In contrast, under the same conditions we observe for a fast AP that $v_{cl}$ rises to a maximum of +16.4 mV, followed by a decrease to a minimum of -8.2 mV. As a consequence, the peak calcium current was delayed by 21 μs and reduced by 19% (-2.0 nA vs -1.7 nA). These effects became even more pronounced at lower $g_{cl}$. The observed delay corresponded to the delay in reaching the transmembrane $v_{pre}-v_{cleft}$ maximum compared to the $v_{pre}$ maximum; the reduction in calcium current was due to a smaller effective AP, which reduced the maximal open probability of the calcium channels. We conclude that developmental acceleration of the calyceal AP will increase the capacitive current to a level that potentially could delay and impede the presynaptic calcium current substantially if $g_{cl}$ were <1 μS.

## Relation between synapse and prespike size

So far, we modeled the cleft as a single, equipotential compartment with capacitive and conductive properties. However, it is expected that the voltage profile within the cleft will depend on synapse geometry [20], and that larger synapses have larger prespikes [30].We therefore need an analytical description of the voltage profile within the cleft. Previous work considered the impact of cleft geometry on the EPSC, a current that runs radially through the cleft and enters at a centered region containing the glutamate receptors [7,20,30]. Our situation is different as we consider a homogeneously distributed current: the capacitive currents arise at every membrane patch facing the cleft. For an analytical description we start by assuming a radially-symmetric synapse with a radius $r$ and a cleft height $h$. Within this synapse we only consider

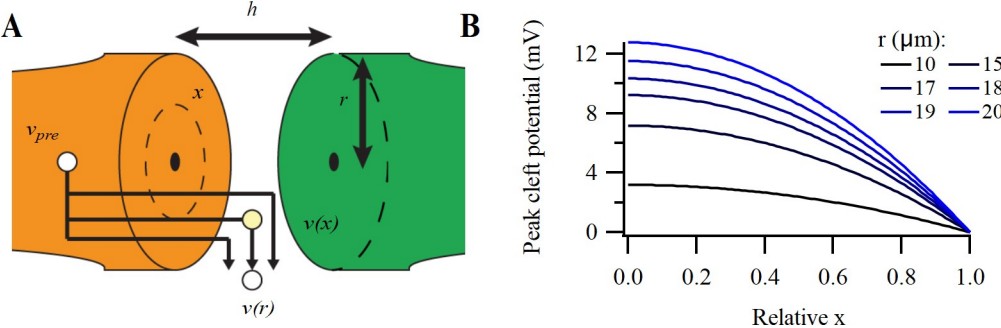

**Fig 7. Synapse geometry determines the profile of the cleft potential.** (A) Drawing of a radially-symmetric synapse with cleft height $h$ and radius $r$. The capacitive currents will generate a cleft potential v(x) at location x. (B) Cleft voltage profile for synapses with different radii (color-coded) at the first peak in the capacitive current. The horizontal axis depicts the location from the center (0) relative to the edge (1.0). In (B), $R_{ex}$ is 100 $\Omega$ cm, $C_m$ is 1 $\mu$F cm$^{-2}$, and h is 30 nm.

capacitive currents from the cleft-facing membrane (Fig 7A). As only a small fraction of these currents enters the postsynaptic cell (the resistive dissipation scenario), we simply assume that all currents run radially to the synaptic membranes. The electric field at location $x$ relative to the cleft center will be $dv_{cl}/dx = xR_{ex}C_m(2h)^{-1}dv_{pre}/dt$ (S2 Appendix B1.3) where $R_{ex}$ is the extracellular cleft resistivity and $C_m$ is the specific membrane capacitance. The electric field grows linearly with the distance from the center. The cleft potential is equal to $\int_x^r dv_{cl}dx$, which gives $v_{cl} = (r^2-x^2)R_{ex}C_m(4h)^{-1}dv_{pre}/dt$ (S2 Appendix B1.4). The size of the cleft potential thus depends on $r^2$, indicating that for larger terminals the cleft potential may become increasingly important (Fig 7B). The voltage profile also shows that not all voltage-gated channels in the cleft will be affected, but only the channels located towards the cleft center.

The temporal changes in the cleft potential will generate a capacitive current at the postsynaptic side. To obtain the VC prespike, we derive $i_{post} = r^4\pi R_{ex}C_m^2(8h)^{-1}d^2v_{pre}/dt^2$ (S2 Appendix B2.2). The VC prespike therefore grows with the fourth power of the radius, which explains why prespikes are only observed in giant synapses such as the calyx of Held synapse. With our geometrical relation for the VC prespike we can relate $g_{cl}$ to extracellular resistivity: $g_{cl} = 8\pi h (R_{ex})^{-1}$ (S2 Appendix B3.2). This relation leads to an estimate for the intra-cleft resistivity of 75 $\Omega$ cm for an $h$ of 30 nm [41] and a $g_{cl}$ of 1 $\mu$S. Considering that there may be extended extracellular spaces within the synaptic cleft [51], and taking into account the variation in $g_{cl}$ (1–2 $\mu$S), we estimate an $R_{ex}$ in the range of 50–100 $\Omega$ cm.

## Fenestration of the calyx of Held reduces cleft potentials

Giant synapses typically fenestrate during development [52–55]. The relation between fenestration and the coupling between pre- and postsynaptic membranes has been studied in numerical simulations [30]. We initially assumed that fenestration minimizes $v_{cl}$ by reducing $g_{cl}$ through reducing the path length to the extracellular fluid ($r$). However, we analytically derived that for a radially-symmetric synapse, $g_{cl}$ does not depend on its radius. The relation between the conductance and the resistivity is $g_{cl} \sim A (r R_{ex})^{-1}$, where $A$ is the area through which the current escapes. As $A$ of a radially-symmetric cleft is $2\pi rh$, when $r$ is doubled, $A$ is also doubled, leaving the conductance independent of synapse size. The cup-shape of the young calyx makes the radially-symmetric synapse a reasonable approximation. This also means that our estimate of the resistivity of the synaptic cleft is relevant for smaller synapses with a comparable radially-symmetric shape.

Following a period of fenestration, the calyx of Held becomes, as its name implies, a synapse with extended fingers [31,56]. However, for finger-like synapses with a sheet-like or cylindrical cleft, the geometrical relations are slightly different. For these synapses we assume that the currents only run transversally, *i.e.* perpendicular to the longitudinal axis. The cleft potential then becomes $v(x) = (r^2 - x^2)(2h)^{-1}R_{ex}C_m dv_{pre}/dt$ (S2 Appendix B4.4), where $2r$ equals the width of the sheet (or $r$ the length of the cylinder). As the escape route for the current is more restricted than for a radially-symmetric synapse, its voltage gradient is steeper. However, the diameter of a calyceal finger will be much less than the diameter of the juvenile calyx, effectively reducing the cleft potential. For a calyceal finger of 4 μm diameter [31,57] the cleft potential at the center of the finger-like synapse would be only ~2.5% of the cleft potential in the cup-shape calyceal synapse (S2 Appendix B5.2). As a consequence, the VC prespike would become ~3.3% assuming that the total contact surface is conserved (S2 Appendix B5.4). The cleft potential and VC prespike with more complex synapse morphology may be investigated using finite-element modeling, but this is beyond the scope of this study.

Fenestration will effectively reduce the amplitude of the cleft potentials, and therefore we expect that the prespike in the adult calyx of Held synapse should be significantly smaller than the juvenile prespike. To study the adult prespike, we revisited *in vivo* whole-cell current-clamp recordings of MNTB principal neurons from wild-type C57BL/6J mice previously reported in refs. [58,59]. CC prespikes consisted of a small positive deflection, preceding the start of the rapid rising phase of the EPSP by about 0.3 ms. In 11 of 12 recordings, the amplitude of the positive deflection was <0.2 mV. In contrast to the juvenile recordings (Fig 1), following the positive peak the membrane potential did not undershoot the baseline and the negative peak was much less conspicuous. An example is shown in Fig 8. The good correlation between the size of CC and VC prespikes (Fig 1C) in combination with the absence of a large decrease in the input resistance of adult principal neurons [58,59] suggests that the VC prespike can be expected to also be relatively small. Our anecdotal evidence from *in vivo* whole-cell recordings in adult mice therefore confirms that the prespike is smaller than the juvenile prespike and that it possibly has a different shape (*cf*. Figs 1 and 8).

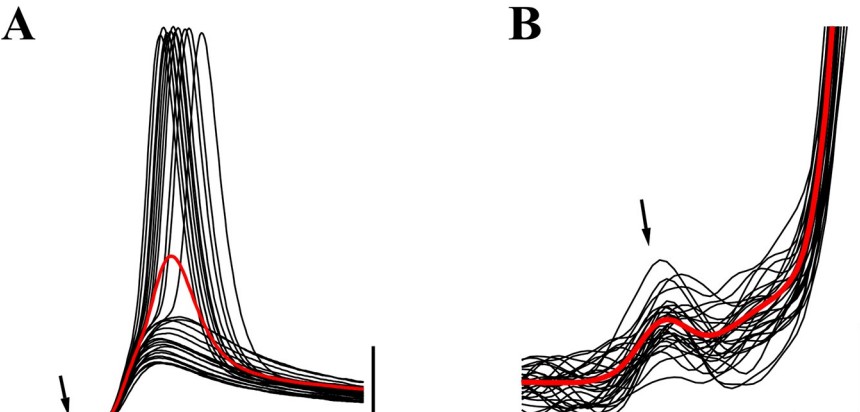

**Fig 8. Adult prespike.** (A) Black traces: Spontaneous, sub- and suprathreshold events recorded during a whole-cell *in vivo* recording of a principal neuron in the MNTB from a P52 C57BL/6J mouse [58]. Events were aligned on the start of their EPSP and (<1 mV) differences in the resting membrane potential between events were subtracted for the display. Minimum interval between events was 20 ms. Red trace: average of >1000 events. Resting potential was -64 mV. Arrow points at the small prespike. Vertical scale bar: 10 mV. Horizontal: 1 ms. (B) As (A), showing close-up of the prespike (arrow). Vertical scale bar: 0.2 mV. Horizontal: 0.2 ms.

## Discussion

Here, we studied how electrical currents that are needed to depolarize a presynaptic terminal or trigger transmitter release are dissipated in the synaptic cleft. We derived equations for the ephaptic coupling between the presynaptic terminal and the postsynaptic cell. We found that the capacitive component of the prespike is closely approximated by the second time derivative of the presynaptic action potential. Its size scales with the fourth power of the radius of the synapse, explaining why intracellularly recorded prespikes are uncommon in the CNS. We showed that presynaptic calcium currents can contribute to the prespike and that their contribution is closely approximated by the scaled first derivative of these currents. We used their contribution to infer an estimate of the cleft conductance, which was ~1 μS for the juvenile calyx of Held synapse. We showed that the cleft conductance of a radially-symmetric synapse does not depend on its radius, but is entirely defined by its height and the extracellular resistivity. This led to an estimate of the cleft resistivity that is at most 50% larger than of the interstitial fluid, which means that the cleft conductance is large enough to allow for the large EPSCs of the calyx of Held, but small enough to have a potential impact on the opening of presynaptic voltage-dependent calcium channels. Finally, we showed that developmental fenestration of the calyx of Held constitutes a very effective adaptation to minimize synaptic cleft potentials.

### Electrical properties of the calyx of Held synapse

Based on our electronic model, we discriminated two extreme scenarios, a scenario in which currents entering the cleft dissipate via the membrane capacitance, and one in which they exclusively dissipate via the cleft conductances. We compared the shape of both the passive and the calcium component of the prespike with both scenarios, and found that the resistive dissipation scenario provided a much better prediction for the shape of the prespike, since the passive prespike resembled the inverted, second derivative of the presynaptic AP much better than the inverted, first derivative, whereas the calcium prespike resembled the inverted, first derivative of the presynaptic calcium currents much better than the inverted calcium currents themselves. There are several other compelling arguments for this scenario. For the calyx of Held synapse to be able to function as a relay synapse in the auditory system, it needs to be both fast and precise, which means a fast presynaptic AP and a large size with many release sites and large EPSCs. We derived for the capacitive dissipation scenario that the size of the prespike would be about half of the size of the presynaptic currents entering the cleft during the presynaptic AP. Because of the large size of the calyx of Held and the fast time course of the presynaptic AP, these currents are in the nA range. The observed size of the VC prespike is therefore clearly not in line with the predictions of the capacitive dissipation scenario. Secondly, the calyceal EPSC is very fast and can easily be 10 nA [36,50,60–62]. This EPSC should flow not only through the postsynaptic glutamate receptor channels, but also through $g_{cl}$ [7]. For the synapse to be able to conduct an EPSC of 10 nA at a driving force of 100 mV, the cleft leak conductance must be at least >0.1 μS. This is consistent with the observed range for $g_{cl}$. These arguments are therefore in line with our conclusion that the resistive dissipation scenario provides a much better description than the capacitive dissipation scenario. Less obviously, the model allowed us to use the calcium component of the prespike to obtain an estimate of the cleft conductance of about 1 μS. This value is subject to some uncertainty. For the calculation we assumed that most, but not all, presynaptic calcium channels are facing the synaptic cleft. The juvenile calyx contains N- and R-channels, which are not located near the release sites based on immunostaining and their lower efficacy at triggering release, which is in contrast to the P/Q-type calcium channels [46]. Our assumption that only two-third of the

calcium current originated from the cleft is therefore reasonable, considering that the N- and R-channels make up about half of the calcium currents of the juvenile calyx.

Even though we calculated the AP-related capacitive currents that run through the cleft-facing membrane, we did not record the AP itself at the cleft-facing membrane. The AP passively invades the terminal [47], and therefore may be slowed down and have a reduced amplitude at the cleft-facing membrane. We consider it unlikely that this filtering is significant for two reasons. First, passive filtering from one end to the other end of the calyx reduces the AP peak by only 6 mV with a delay of <0.2 ms [63]. Second, the axonal AP is able to invade the fenestrated endings of the mature calyx [64], which should have a significantly higher impedance than the cup-shaped calyx. Our calculation of the presynaptic capacitive currents as the rate of change of the recorded calyceal AP relative to the cleft potential is therefore likely to be accurate.

## Comparison of cleft electrical properties across synapses

Interestingly, with similar values for cleft height and extracellular resistivity other studies came to a wide range of estimated values for the cleft conductance, both at least an order of magnitude smaller than 1 μS [11,26,28–30,65], but in some cases also considerably larger than 1 μS [7,66]. In some of the synapses, an unusually large and narrow path to the interstitial fluid was responsible for the low estimates. Examples are mossy fiber boutons penetrated by a thin (<1 μm) spinule [28], the unusually long calyx-type vestibular synapse [11], or the cone-to-horizontal cell synapse, for which a large part of the horizontal cell is located within an invagination of the cone photoreceptors [67]. While cleft resistivity should be independent of morphology, the cleft conductance for membrane currents at the deep end of these synapses would be significantly less than for the homogeneously distributed current addressed here. In these morphologically complex synapses, slow removal of ions or neurotransmitter may serve a function, for example in the case of vestibular hair cells, where potassium ions accumulate [10,11], or in the cone photoreceptor synapse, where prolonged effects of glutamate are important, and where postsynaptic currents are typically relatively small [68]. At the other end of the range lies the high estimate for the cleft conductance of the neuromuscular junction, to which the postjunctional folds were suggested to contribute [7]. The unique morphological properties of these synapses have therefore made a large contribution to the large range of estimated values for the cleft conductance.

Our calculations indicated that for homogeneously-distributed currents the associated cleft conductance is fully determined by the extracellular resistivity, the cleft height and the morphological type of the synapse, but not by the size of the synapse. The cleft height is constrained within narrow boundaries because of the opposing requirements of low volume for high neurotransmitter concentrations and high conductance to allow large synaptic currents [16]. The cleft height typically ranges across synapses between 10–30 nm, but there is some uncertainty regarding the impact of chemical fixation. Assuming a cleft height of 30 nm, we estimated a resistivity of 75 Ω cm based on the prespike, only about 30% larger than of cerebrospinal fluid or Ringer solution [16,69]. While we are not aware of other estimates of the extracellular resistivity within the synaptic cleft, our estimate is similar to the intracellular resistivity of neurons [70–72] and the periaxonal resistivity of myelinated axons [72]. The extracellular resistivity can also be estimated based on the diffusional mobility of small molecules. Our estimate agrees with those made with an imaging method in hippocampal mossy fiber synapses, which also estimated that small molecules move on average about 50% slower than in the Ringer solution [73]. Estimates of neurotransmitter diffusion constants within the synaptic cleft range from 1.5 to 4 times smaller than in free solution [74–76], but for neurotransmitters diffusion within the synaptic cleft may be slowed down by neurotransmitter-binding proteins. We therefore suggest that a cleft resistivity of ~75 Ω cm may apply to other synapses as well. Given the range of

cleft heights and the estimated cleft resistivity, the cleft conductance can be inferred for a distributed current using our analytical relations (B3.2). It should be not so different from the calyx of Held considering that cleft height is conserved across synapses, while the geometry of the juvenile calyx translates well to bouton synapses by simple downscaling. Cleft conductances for circular synapses would then range between 0.3–1.0 µS.

## The prespike: what is it good for?

The extracellular AP of the calyx of Held can be observed in extracellular recordings at the target neuron [77], and the characteristic shape of the complex waveform consisting of the presynaptic AP, the EPSP and the postsynaptic AP has facilitated identification of *in vivo* single unit recordings at the MNTB [59]. Importantly, it was shown that the calyceal AP can be picked up in postsynaptic whole-cell recordings as well [34]. This extracellular field potential outside the cleft cannot generate a 0.1 nA-prespike amplitude intracellularly without assuming an obviously non-physiological postsynaptic admittance (~0.2 µS). Incidentally, this also indicates that other compartments, such as the presynaptic heminode, contribute very little to the generation of the prespike. In postsynaptic voltage clamp recordings the prespike showed a resemblance to a high-pass filtered version of the presynaptic AP [36], but we showed here that it is closer to the second derivative of the AP. At the related endbulb of Held synapse, the presynaptic AP is also readily picked up in extracellular recordings [78]. However, no prespikes were reported during postsynaptic whole-cell recordings [79] possibly due to the smaller size of the endbulb compared to the calyx of Held. In the visual system, prespikes can be seen in a subset of lateral geniculate neurons [80]. However, in many synapses in which the presynaptic AP is readily observed in extracellular recordings, it is not reported in intracellular postsynaptic recordings [18,81–85], attesting to the efficiency of the synaptic cleft in dissipating these currents. Interestingly, especially in the hippocampus, neurons can display so-called spikelets with an amplitude of mV in intracellular recordings. They have a biphasic shape resembling the calyceal prespike recorded in current-clamp. These spikelets are thought to result from ephaptic, capacitive coupling with one or more nearby neurons [86]. The equations we derived here should be applicable to these spikelets as well: we predict them to resemble the second derivative of the AP in the nearby neurons [87], and to critically depend on contact area, but also on proximity and on the resistivity of the extracellular fluid.

We believe that the major reason for the lack of prespikes at most synapses is that the prespike amplitude is proportional to the 4th power of the synapse radius (S2 Appendix B2). During the postnatal development of the MNTB, it can therefore be argued that the absence of postsynaptic recordings with multiple prespikes cannot be used as firm evidence against the presence of multiple calyces on these cells. We previously found anecdotal evidence for the presence of multiple large inputs during early postnatal development [88]. In a small fraction of the recordings multiple large inputs were observed, one with and one without a prespike. It might be that the one without the prespike had a smaller surface area or a higher cleft height, going below the radar. This highly nonlinear dependence on apposition area may thus provide an explanation for the discrepancy between anatomical evidence favoring multiple large inputs during development [89] and the lack of recordings with multiple prespikes.

## Divide and conquer

Virtually all synapses have in common that neurotransmission is initiated by a presynaptic action potential that results in capacitive cleft currents. The size of these currents depends on the amplitude and speed of the presynaptic AP. The properties of the presynaptic APs have mostly been

studied at giant synapses, where they tend to be large and fast [50,90–95]. These giant synapses are typically relay synapses that are specialized for rapid signaling, but recent measurements at more conventional bouton synapses in the neocortex show that their presynaptic action potentials are large and fast as well [96]. These capacitive currents, together with the presynaptic ion channel currents, will lead to a change in the cleft potential whose time course will closely follow the time course of the presynaptic membrane currents, and whose amplitude will critically depend on the cleft resistivity. Because of this cleft potential, the presynaptic ion channels facing the cleft will sense a reduced action potential. We showed that this should result in a less effective and delayed opening of the presynaptic calcium channels facing the synaptic cleft. Since rapid and large changes in the voltage sensed by the presynaptic ion channels serve to rapidly turn transmitter release on and off again, large cleft potentials seem undesirable, even though they may serve other, more beneficial functions such as influencing the neurotransmitter mobility within the cleft [15,16], or rapidly inhibiting postsynaptic action potential generation [18,97].

Above, we argued that many radial-like synapses should have a cleft conductance similar to what was measured here. An important difference between small and large synapses, however, is that the total current will be much larger in the large synapse, as the capacitive current scales with the presynaptic apposition area. A large synapse thus has to dissipate this much larger current over a cleft conductance that is similar to the cleft conductance of a small synapse. There are three obvious different neural strategies to mitigate significant cleft potentials. Firstly, a strategy may be widenings of the synaptic cleft in between synaptic junctions. These intrasynaptic space enlargements [4], also called channels of extracellular spaces [98,99] or subsynaptic cavities [100], have been found at many synapses, including the calyx of Held synapse, and may increase cleft conductance. Interestingly, a knockout of brevican, a component of the extracellular matrix of perineuronal nets, leads to a reduction of the size of the subsynaptic cavities at the calyx of Held synapse and an increased synaptic delay, even though the relation with a change in cleft conductance remains to be established [100]. The second strategy is to divide the cleft membrane currents over many postsynaptic junctions. For example, in the case of the squid giant synapse, the postsynaptic axon has numerous processes contacting the presynaptic axon, each with a width that is only ~1 μm [101]. At each of these processes the cleft potential will be small, thus effectively mitigating its impact. The last strategy is to fenestrate presynaptically. Most large synapses fenestrate presynaptically during development, leading to extended extracellular spaces [99]. Well-known examples include the neuromuscular junction, where the terminal arbor assumes a pretzel shape [102], the calyx of Held, which develops from a cup shape to the calyx shape [52,53], the endbulb of Held synapse [54] or the avian ciliary ganglion [55]. In S2 Appendix B5 we quantified the impact this will have on the size of the cleft potential change and on the prespike. Previously, other roles for these adaptations have been emphasized, such as speeding up of transmitter clearance or minimization of crosstalk between postsynaptic densities, both for the calyx of Held synapse [52,103,104] and at other synapses [105,106]. The digitated adult morphology of the calyx of Held is accompanied by an increase in the number of puncta adhaerentia [104] and the formation of swellings connected via necks to the fingers [42,51]. In the adult calyx of Held, a large fraction of the release takes place at these swellings, which should facilitate the dissipation of capacitive currents by the cleft, and reduce the impact of the cleft conductance on the postsynaptic currents.

In conclusion, we show here that the prespike of the calyx of Held synapse can be used to infer critical properties of its synaptic cleft; apart from the cleft conductance, it may also provide information about the presence or absence of ion channels at the cleft, or the effectiveness of morphological changes in reducing the size of cleft potentials during development. We thus provide a quantitative description for how the synaptic cleft, despite its narrow, limited space, can sustain the large and fast currents that are at the heart of chemical neurotransmission.

## Materials & Methods

The AP template shown in Fig 4A was based on an AP template previously used in voltage-clamp experiments [43].

For the experimental procedures and recording conditions for the dual recordings of the calcium current and the postsynaptic currents, we refer the reader to the original paper in which these recordings were reported [44]. For the current paper, we used trains of 45–50 identical APWs at an interval of 100 Hz. These APWs were what was called the 'control AP' in ref. [44], i.e. the first AP of a train of presynaptic APs. To limit the possible impact of EPSCs on the prespike during the trains, the first 20 APWs were skipped, and the responses to only the next 25 APWs were averaged. The presynaptic passive (capacitive) component was estimated by presenting the same train five times at 5x smaller amplitude, and summing the responses (P/5 method). The capacitive component of the prespike was isolated analogously from the same reduced APW train. *In vivo* whole-cell current-clamp recordings of developing MNTB principal neurons with afferent stimulation were previously reported in ref. [37] (*n* = 6). *In vivo* whole-cell current-clamp recordings of MNTB principal neurons from adult C57BL/6J mice were previously reported in ref. [58] (*n* = 7, only wild-type animals) or in ref. [59] (*n* = 5). We also refer to these papers for the respective ethics statements.

For numerical simulations with voltage-gated channels the following two differential equations were defined, following [30]:

$$c_{post} \frac{dv_{post}}{dt} = c_{cl} \frac{dv_{cl}}{dt} - g_{post}(v_{post} - v_{rest})$$

$$2c_{cl} \frac{dv_{cl}}{dt} = c_{cl}\left(\frac{dv_{pre}}{dt} + \frac{dv_{post}}{dt}\right) - g_{cl}v_{cl} + \sum g_{max}p_{open}(v_{pre} - v_{cl} - v_{channel})$$

Where $g_{max}$ represents the maximal conductance of a voltage-gated channel, $p_{open}$ the open probability of the voltage-gated channels, which is calculated using a multiple-state model (see below), and $v_{channel}$ the reversal potential of the voltage-gated channel. The reversal potentials were 50 mV, -90 mV, 40 mV and -70 mV for the sodium conductance, potassium conductance, calcium conductance, and the resting membrane potential of the postsynaptic neuron, respectively. $v_{pre}$ was defined by an AP waveform. Conductance densities in Fig 5 were 0.2 nS $\mu$m$^{-2}$ for the sodium conductance, 0.08 nS $\mu$m$^{-2}$ for the high-threshold potassium conductance, 0.24 nS $\mu$m$^{-2}$ for the low-threshold potassium conductance, and 0.04 nS $\mu$m$^{-2}$ for the calcium conductance with a cleft-facing membrane area of 1000 $\mu$m$^2$.

Voltage-gated conductances were modeled with multiple states following ref. [107] except for the calcium conductance. In the models C indicates a closed state, O the open state and I the inactivated state. The sodium channel followed a five states-model, the high-threshold and low-threshold potassium channel were modeled as a six-states channel, and the calcium channel as a two-state channel for which the open probability was squared:

$$\text{Five states-model}: \quad C_1 \underset{\beta}{\overset{2\alpha}{\rightleftharpoons}} C_2 \underset{2\beta}{\overset{\alpha}{\rightleftharpoons}} C_3 \underset{\delta}{\overset{\gamma}{\rightleftharpoons}} O \overset{\theta}{\underset{\epsilon}{\rightleftharpoons}} I$$

$$\text{Six states-model}: \quad C_1 \underset{\beta}{\overset{4\alpha}{\rightleftharpoons}} C_2 \underset{2\beta}{\overset{3\alpha}{\rightleftharpoons}} C_3 \underset{3\beta}{\overset{2\alpha}{\rightleftharpoons}} C_4 \underset{4\beta}{\overset{\alpha}{\rightleftharpoons}} C_5 \underset{\delta}{\overset{\gamma}{\rightleftharpoons}} O$$

$$\text{Two states-model}: \quad C \underset{\beta}{\overset{\alpha}{\rightleftharpoons}} O$$

The rate constants indicated by the Greek letters in the state models were calculated as

$$k = a \, e^{b(v_{pre} - v_{cl})}$$

**Table 1. Rate constants for the voltage-gated channels.**

| | | a (ms$^{-1}$) | b (mV$^{-1}$) |
|---|---|---|---|
| Sodium conductance | α | 48.9 | 1/128.4 |
| | β | 1.46 | -1/8.9 |
| | γ | 392.8 | 1/36.0 |
| | δ | 0.73 | -1/15.5 |
| | ε | 7.8 | 1/35.9 |
| | θ | 0.01 | -1/18 |
| High-threshold potassium conductance | α | 1.097 | 1/57.404 |
| | β | 0.794 | -1/79.264 |
| | γ | 33.750 | 0 |
| | δ | 74.360 | 0 |
| Low-threshold potassium conductance | α | 1.204 | 1/37.574 |
| | β | 0.360 | -1/230 |
| | γ | 245.488 | 0 |
| | δ | 132.566 | 0 |
| Calcium conductance | α | 1.78 | 1/23.3 |
| | β | 0.14 | -1/15 |

Parameters were taken from ref. [107], except for the calcium conductance [45].

where k represents a particular rate constant, and *a* and *b* are channel-specific parameters listed in Table 1. To take into account a possible effect of the cleft potential on the channel kinetics, the rate constant was calculated using the voltage difference between the presynaptic potential and the cleft potential.

The ordinary differential equations were numerically solved in Igor using IntegrateODE with default settings (fifth-order Runga-Kutta-Fehlberg; step size was adapted to keep $Error_i < 10^{-6}$; $Error_i < 10^{-10}$ did not lead to substantial differences in the prespike waveforms).

Statistics are reported as mean and standard deviation. Pearson's correlation coefficient was calculated to quantify correlations. Paired t-tests were performed to compare correlation coefficients from the same data and to compare the absolute deviation from the origin.

## Supporting information

**S1 Appendix. A1 Analysis of the electronic model & A2 The calcium prespike.** In this Appendix we provide, given the electronic model, the analytical relations for the cleft potential, the VC prespike and the CC prespike, and we address how the presynaptic calcium current couples to the postsynaptic recording.
(PDF)

**S2 Appendix. B Geometry.** In this appendix we provide the geometrical relation for the cleft voltage profile for a radially-symmetric synapse as a result of the capacitive currents from the presynaptic action potential, the postsynaptic current this will generate, how the cleft conductance relates to the extracellular resistivity assuming a radially-symmetric synapse, what these relations become for a finger-like synapse, and how the transformation from a radially-symmetric synapse to synapse composed of multiple fingers changes the cleft potential profile.
(PDF)

**S1 Fig. Relation between the calcium current and the prespike, continued.** Our data set was composed of five double recordings. The presynaptic voltage-clamp waveform is shown in Fig

4A. (A) *from top to bottom*: The presynaptic calcium current (after P/5 subtraction), its inverted first derivative, the calcium prespike, and the active (black, thick) and passive prespike (magenta, P/5-scaled). Voltage-gated sodium and potassium currents were blocked (see Materials and Methods). The left-most column is also shown in Fig 4A. The right-most is the only one plotted on scaling of the axes on the right. Notice how even when the presynaptic calcium current does not smoothly follow a Gaussian function, its first derivative resembles the calcium prespike as predicted from our analytical relations for the resistive dissipation scenario. (B) Relation between the inverted first derivative of the presynaptic calcium current and the calcium prespike (black dots). Maximum correlation was obtained by introducing a delay for the first derivative. This delay and the slope of the regression line are shown in the top left corner of each panel. Each panel corresponds the example in the same column in A.
(TIF)

## Acknowledgments

We thank Dr. Henrique von Gersdorff for helpful discussions on the developmental changes in the prespike.

## Author Contributions

**Conceptualization:** Martijn C. Sierksma, J. Gerard G. Borst.

**Data curation:** J. Gerard G. Borst.

**Formal analysis:** Martijn C. Sierksma.

**Investigation:** J. Gerard G. Borst.

**Methodology:** Martijn C. Sierksma.

**Resources:** J. Gerard G. Borst.

**Software:** Martijn C. Sierksma, J. Gerard G. Borst.

**Supervision:** J. Gerard G. Borst.

**Validation:** Martijn C. Sierksma, J. Gerard G. Borst.

**Visualization:** Martijn C. Sierksma, J. Gerard G. Borst.

**Writing – original draft:** Martijn C. Sierksma, J. Gerard G. Borst.

**Writing – review & editing:** J. Gerard G. Borst.

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
