## [Decision Letter · Decision Letter 0]

22 Jul 2021

Dear Dr. Borst,

Thank you very much for submitting your manuscript "Using ephaptic coupling to estimate the synaptic cleft resistivity of the calyx of Held synapse" for consideration at PLOS Computational Biology.

As with all papers reviewed by the journal, your manuscript was reviewed by members of the editorial board and by several independent reviewers. In light of the reviews (below this email), we would like to invite the resubmission of a significantly-revised version that takes into account the reviewers' comments.

We cannot make any decision about publication until we have seen the revised manuscript and your response to the reviewers' comments. Your revised manuscript is also likely to be sent to reviewers for further evaluation.

Sincerely,

Hugues Berry

Associate Editor

PLOS Computational Biology

Daniele Marinazzo

Deputy Editor

PLOS Computational Biology

Reviewer's Responses to Questions

**Comments to the Authors:**

Reviewer #1: In the manuscript by Sierksma and Borst, the authors addressed a phenomenon of ephatic coupling between the pre- and postsynaptic neurons that occurs due to a capacitive transmission of the voltage transients between cells and due to a finite conductance of the synaptic cleft dissipating the spatially distributed currents generated by the opposing membranes. This effect is most prominent at the synapses with large surface area and large currents, therefore a specialized auditory synapse, the calyx of Held, at studying which the author’s laboratory has renowned expertise, was used as a model system to validate and further explore the theoretical outcomes.

Using a phasor-based analysis approach to solve the equivalent circuit of the synapse that incorporated cleft resistance and cleft capacitance, the authors derived a compact analytical solution for postsynaptic prespike well measurable at MNTB principal cells in either current-clamp or voltage-clamp modes. The derived solution predicted that a “passive” component of the prespike current recorded in voltage-clamp should be proportional to the inverted second derivative of the presynaptic AP transient, and an “active” contribution from the presynaptic calcium current – proportional to the inverted first derivative of that current. The authors used recordings from their previously published datasets to validate the predictions of phasor analysis, which worked out well, and further used the analysis to estimate the cleft conductance. Next, numerical simulations were used to model a possible contribution of sodium and potassium voltage-gated currents to the prespike, however due to a mismatch with the experimental data it was concluded that sodium and potassium channels were largely excluded from the synaptic cleft. Finally, using numerical simulations, the authors demonstrated that accelerated “mature” AP waveform would have resulted in increased cleft potential leading to suppression of AP-driven gating of the presynaptic Ca current, if the synapse geometry did not change during maturation. Developmental fenestration of the presynaptic terminal was proposed as a natural mechanism to reduce the cleft potential, and quantitative estimates of developmental changes in the cleft parameters were made accordingly.

The manuscript is clearly written, theoretical derivations appear correct, and were scholarly described in the Appendix. The experimental data, even though revisited from previous publications, are of high quality and no further experiments are required within the scope of the study. The conclusions are well justified and discussed. However, despite a high stand-alone quality of the study, there are certain concerns regarding the conceptual novelty and the novelty of some conclusions (see major points below). Given these points can be addressed, it could render the manuscript suitable for publication after a round of revision that is not expected to take long.

Major points:

- As the authors well acknowledged in Introduction, the phenomenon of ephatic coupling has been previously addressed by a number of studies. Here, a novel way was devised to solve the circuit (Figure 2B) using phasors. However, as the authors also admitted, a compact analytical solution with phasors is only possible under certain simplifications (Y1 and Y4<<y2 and="">>4*pi*f*Ccl; see Appendix) which effectively reduced the model circuit to a single-compartment configuration. While in reality, the proximal axonal heminode containing sodium channels (Leão et al 2005 J Neurosci, not cited) connected to a “passive” terminal with extrasynaptic and cleft-opposing membranes facing different transmembrane voltages, together assume at least two (or rather three) functional compartments (see Savtchenko 2007, ref. 30; also apparent from the results in this manuscript, Figure 5). On the one hand, the phasor predictions result in elegant expressions and agreed well with experimental data, but on the other hand this solution is valid only for particular cases like capacitively transmitted “passive” prespike and the “active” contribution of symmetric calcium current. For other, very important and interesting questions (Figures 5, 6) the authors had to switch to numerical simulations solving the system of coupled differential equations as it was done previously (e.g. ref. 30). The authors should better elaborate on the advantage of introducing phasors to this problem as compared with the approaches used previously, and state whether it was/was not possible to infer the electrical properties of the synaptic cleft using previously existing solutions.

- Based on the computed prespike waveform (Figure 5), the authors noted that homogeneous distribution of Na-channels (VGSCs) in the calyx of Held terminal was not compatible with the experimental observations and thus concluded that VGSCs are excluded from the cleft opposing membrane. Furthermore, the prespike analysis was proposed as a method to analyze the presence of different conductances at a synapse (lines 310-314). However, the authors did not cite a study (Leão et al 2005 J Neurosci) demonstrating the absence (at least a low expression level) of VGSCs at the terminal, which should be introduced to support the computational outcome. In addition, the authors could extrapolate to the situation when VGSCs are present (e.g. at hippocampal mossy fiber boutons, Engel and Jonas 2005, Neuron; not cited) and possibly provide a reference to a study that documented a prespike waveform with a sharp spikelet contribution from Ina at such synapses (if such evidence is available in the literature).

- The authors claimed (lines 75-77) that the prespike “is probably due to ephaptic coupling of the presynaptic AP via the synaptic cleft [30, 35], but the precise relation between the presynaptic AP and the prespike has not been studied“. However, the study by Savtchenko 2007 (ref. 30) clearly showed the waveform of a capacitive prespike (Figure 3A therein) and its dependence on synapse radius and resistivity. Moreover, the same study (ref. 30) has discussed the effect of developmental fenestration of the calyx of Held synapse on the reduced ephatic coupling due to reduction of the cleft potential, with geometrical considerations very similar to those made in the last part of the manuscript. Thus, it seems necessary to provide more credit to previous studies (in particular, ref. 30) when modelling the effect of developmental fenestration on the cleft potential (pages 19-22).

Minor:

- Figure 1 legend, line 96 (“Note the small upward deflection…”), and line 309 (“upward inflection in the CC prespike and such an inflection can sometimes be observed”): for clarity, please mark these inflections with arrows on the Figures 1D and 5B.

- Please add an annotation “ica” in the Figure panel 3F, for compatibility with other panels.

- Line 179: “Figure 2F” should read “Figure 3F”.

- Line 221: “we averaged the postsynaptic responses to the P/5 AP waveform” – please indicate the amplitude of unitary responses to P/5 AP waveforms (was it feasible to measure?). Did it correspond to a prediction made by A2.6, i.e. few tens of pA?

- Figures 4 and 6: please choose better contrasting pairs of colors (blue vs black are very hard to discriminate even on a screen)

- In the appendix A, please briefly introduce the meaning of atan2() function for a broader readership

- Lines 938 (A1.5) and 939: for consistency, please use notation “gcl” instead of “g3”

- Line 970: a closing bracket is missing in denominator

- Line 604: in Methods, please explicitly specify the default parameters of IntegrateODE solver (Runge-Kutta which order, integration step selection) – these parameters do matter for integration over rapidly changing transients like Na-current

- Unformatted references found on lines 49, 63, 400, 401, 578, 579, 580, 591</y2>

Reviewer #2: In this study, Sierksma and Borst take an effort to quantitatively describe the ephatic coupling between pre- and postsynaptic compartments of the calyx of Held. The authors derive a theoretical framework to predict postsynaptic membrane potential changes and postsynaptic currents caused by ephatic coupling of presynaptic action potentials that precede the synaptic, neurotransmitter-mediated response (prespike). The theory is sound and relies on the assumption of a leak conductance along the synaptic cleft and a sequential arrangement of capacitors towards the postsynaptic cell. The equations derived in this study are used to simulate postsynaptic responses which are contrasted to previously published experimental data. The authors propose a relevance for presynaptic Ca2+ currents for the shape of the signals. The predictions indeed bear resemblance of experimental recordings and the authors can estimate the extracellular cleft resistivity. The authors furthermore consider how geometrical properties impact this and the prespike.

This is a thorough and rigorous analysis with high level of detail regarding the theoretical derivation of equations to calculate responses from equivalent circuits. Though the study focusses on a re-analysis of previous experimental data, the developed theory and analysis are novel in the sense that they can provide a framework to explain prespikes that have been observed at Calyx of Held type synapses under certain conditions. However, though the example traces show agreement, the study falls short in providing a systematic and quantitative comparison of the predicted responses with the experiment, making it difficult to assess how well the theory overall describes the observation and how robust the phenomenon itself is (i.e. how stable the parameter values are). The observation of prespikes is limited to few specialized synapse types with large surface area (such as the immature Calyx of Held described here) and it does not become clear whether and how the conclusions of this study can be generalized to other (i.e. the majority of) synapses and what the functional relevance of the prespikes may be. The authors provide some interesting hypotheses of putative functional relevance at the calyx in their discussion, but the study would benefit greatly if these were developed further and evaluated in the context of the determined parameters towards direct predictions of the impact on synaptic function under physiological conditions. Thus, while the study will surely be of interest to specialists, it currently does not address a broader audience.

Major

1. A more quantitative evaluation of the theory is needed. In the section entitled “Experimental validation of the model” the authors show comparisons of their theoretical prediction of the prespike compared to experimental measurements. The authors state that “The prespike generally matched the scaled inverted second derivative of the AP”, but only one example trace is shown here and the overall congruence between the traces in panel A is not too striking (by optical inspection). The authors provide some more quantitative comparison in panel B, which is good, but the measure (peak-peak) has rather limited capacity to describe the overall shape of responses as it heavily relies on the presence/detection of the negative-going peak. Here it would be necessary to see more examples and e.g. fit the parameters for each trace by minimizing the deviation between simulation and experimental traces (using e.g. a Chi² value). Is optimal overlap achieved for a unique set of parameters? A comparison of Chi2 values for the two models (scaled/shifted) AP’ and AP’’ might provide an even clearer picture as to how superior the proposed theory is. Moreover, it would be informative to see a systematic quantification of the best fit parameters/scaling factors/phase shift for all recordings.

2. The study would benefit if its descriptive nature could be extended to use the theory to make a specific, quantitative prediction that could either be tested experimentally in the future or be used to investigate novel aspects in existing datasets. Some suggestions might be (but are not limited to) the following: One interesting aspect is the simulation of AP-induced presynaptic Ca2+ currents and their dependence on the cleft’s resistivity, but this remains rather preliminary. Which type of physiological stimuli might be impacted and how? The study following brevican deletion cited in the discussion appears to back up the author’s prediction, but is there a more quantitative argument that can be made (e.g. taking into account the change in cleft width)? Maybe the authors’ theory can be used to make some predictions regarding the bi-directional nature of the ephatic coupling (using the estimated parameters to predict a putative presynaptic effect by postsynaptic receptor activation)?

Minor

1. Indicate (e.g.) by arrows in Fig. 1A, B where the signals are measured that are shown in Fig. 1C.

2. It is stated that one assumption is that the capacitance facing the cleft is equal to the postsynaptic one. This rationale should be explained further.

3. Can the authors explain how non-perfect postsynaptic capacitance compensation in the VC experiment might affect the estimation?

4. Please indicate how Ca2+ currents were blocked in the experiment depicted in Fig. 3 directly in the figure legend to make it easier for the reader to obtain this information.

Reviewer #3: This MS analyses the effects of the presynaptic action potential at the calyx of Held on the voltage in the synaptic cleft by exploiting postsynaptic recordings, in which, under some conditions at least, a signal reflecting the presynaptic action potential can be observed. The authors construct a plausible equivalant circuit of the calyx, enabling them to identify the fundamental mechanisms governing these processes. In particular, they conclude that the peak voltage in the cleft scales with the 4th power of its radius. In addition, the authors show that the presynaptic calcium current can also be investigated via its effect on the cleft potential. The authors are also able to account for the effects of the morphological changes occurring during development.

Overall, I found this paper to be original, carefully constructed and interesting. In my opinion the modelling is quite judicious, with an informative level of approximation and a thorough exploitation of available data. I particularly enjoyed the ability to analyse the calcium current.

I have one major concern, which is that in trying to check the equations of the modelling, I fell at the first fence. I was unable to verify equation A1.1 (in the Appendix). Specifically, in my attempt I obtain

V_cleft/V_pre = jY_2/(Y_3 + j(Y_2 + Y_5))

i.e., without the minus signs. I don't believe the two versions to be equivalent. Maybe the authors are using a non-standard version of complex impedance?

Obviously, I would like the authors to check this equation and, if there is indeed an error, investigate how it affects the rest of the modelling/analysis. I haven't attempted to calculate how great an effect the apparent error would have, but the equation is quite fundamental to the remainder of the paper. The authors may find it beneficial to employ explicit circuit modelling software (such as Spice; I believe free versions are available) to provide some additional assurance that their analytical solutions are correct.

Despite this potential error, I still find the whole approach original and interesting and I look forward to discovering whether I am mistaken, whether none of the conclusions are materially affected or whether different conclusions pertain.

At this stage I have few other comments. As a matter of personal preference, I would find the figures more readable with additional labelling, indicating for instance the nature of the traces in Fig. 1A,B and Fig. 4A,C.

**Have the authors made all data and (if applicable) computational code underlying the findings in their manuscript fully available?**

Reviewer #1: **No: **The experimental data were recycled from the previous publications co-authored by Dr. Gerard Borst. These raw data underlying traces in Figures 1, 4, 8 were, to my knowledge, not made publicly available (the data in Figures 1 and 4 being more crucial for the statements made in the manuscript). I believe, making these data available could be easily done.

The IgorPro code for numerical simulations of the sodium, potassium and calcium current contributions to the cleft potential and to the postsynaptic and presynaptic currents (Figures 5, 6) was not made available, but this might not be necessary as the underlying equations and the simulation parameters were well documented.

Reviewer #2: Yes

Reviewer #3: Yes

PLOS authors have the option to publish the peer review history of their article (what does this mean?). If published, this will include your full peer review and any attached files.

Reviewer #1: No

Reviewer #2: No

Reviewer #3: No
---

## [Decision Letter · Decision Letter 1]

5 Oct 2021

Dear Dr. Borst,

We are pleased to inform you that your manuscript 'Using ephaptic coupling to estimate the synaptic cleft resistivity of the calyx of Held synapse' has been provisionally accepted for publication in PLOS Computational Biology.

Best regards,

Hugues Berry

Associate Editor

PLOS Computational Biology

Daniele Marinazzo

Deputy Editor

PLOS Computational Biology

Reviewer's Responses to Questions

**Comments to the Authors:**

Reviewer #1: In the revised manuscript by Sierksma and Borst and in their rebuttal letter, the authors addressed well my points, and as far as I can judge, also the points of other reviewers. I was surprised that the authors decided to omit the nicely developed phasor solution for the problem: it was well done and educational at the same time. But I have to agree, with a more traditional derivation based on Kirchhoff’s laws, the manuscript and derivation in the appendix has become significantly more comprehensive. The new derivations appear correct to me given provided assumptions that were well justified, and importantly, they led to the same outcome as the phasor-based solution which independently confirm their correctness. The updated Figures 3, 4 look better and the discussion reads well.

In summary, at this stage I would recommend to accept this manuscript for publication in PLoS Computational Biology without a need for separate revision.

Some minor proofs kind of changes that I noticed are listed below:

- Line 16: “cell” should be “cells”?

- Line 22: comma should be omitted

- Lines 99-100: please mention what the deflection corresponds to and possibly reference Figure 5

- Lines 126, 134, 136, 144, 151-152: please provide references to the corresponding equations in appendix, replicating or used to derive the formulas in the main text

- Lines 171 & 175: please introduce the term tau_cl nearby these first and second occurrences in the main text, it is only done on the line 209

- Line 171: was “-“ omitted in the equation for i_post?

- Line 190: the second “between” should read as “of”

- Line 218: Appendix A2, not A4.

- Lines 244-245: calcium prespike is plotted against the derivative of calcium current, not vice versa

- Line 246: Figure “S1”, not “9B”

- Line 250: “VC” should read as “prespike”, and “both” should be removed

- Line 326: “for gcl” should read as “of gcl”

- Line 412: “the positive peak was not followed by a negative peak” – this notion is arguable because of the small prespike amplitude and non-stationary EPSP “pre-foot” at that scale (Figure 8B)

- Line 979: the second “and” should read as “are”

- Line 1040: an extra “and”

- Line 1059: A1.6, not A1.9

- Line 1064: A1.16, not A1.19

Reviewer #2: The authors have significantly improved their manuscript with their revisions. I have no further reservations and recommend the publication of the article in its current form.

Reviewer #3: The authors have satisfactorily answered my one major point. I'm glad they are not using non-standard phasors.

**Have the authors made all data and (if applicable) computational code underlying the findings in their manuscript fully available?**

Reviewer #1: Yes

Reviewer #2: None

Reviewer #3: Yes

PLOS authors have the option to publish the peer review history of their article (what does this mean?). If published, this will include your full peer review and any attached files.

Reviewer #1: No

Reviewer #2: No

Reviewer #3: No

---

## [Editor Report · Acceptance letter]

21 Oct 2021

PCOMPBIOL-D-21-00785R1 

Using ephaptic coupling to estimate the synaptic cleft resistivity of the calyx of Held synapse

Dear Dr Borst,

I am pleased to inform you that your manuscript has been formally accepted for publication in PLOS Computational Biology. Your manuscript is now with our production department and you will be notified of the publication date in due course.

With kind regards,

Katalin Szabo
